# ROBUST IMITATION LEARNING FROM CORRUPTED DEMONSTRATIONS

## ABSTRACT

We consider offline Imitation Learning from *corrupted demonstrations* where a constant fraction of data can be noise or even arbitrary outliers. Classical approaches such as Behavior Cloning assumes that demonstrations are collected by an presumably optimal expert, hence may fail drastically when learning from corrupted demonstrations. We propose a novel robust algorithm by minimizing a Median-of-Means (MOM) objective which guarantees the accurate estimation of policy, even in the presence of constant fraction of outliers. Our theoretical analysis shows that our robust method in the corrupted setting enjoys nearly the same error scaling and sample complexity guarantees as the classical Behavior Cloning in the expert demonstration setting. Our experiments on continuous-control benchmarks validate that existing algorithms are fragile under corrupted demonstration while our method exhibits the predicted robustness and effectiveness.

## 1 INTRODUCTION

Recent years have witnessed the success of using autonomous agent to learn and adapt to complex tasks and environments in a range of applications such as playing games (e.g. Mnih et al., 2015; Silver et al., 2018; Vinyals et al., 2019), autonomous driving (e.g. Kendall et al., 2019; Bellemare et al., 2020), robotics (Haarnoja et al., 2017), medical treatment (e.g. Yu et al., 2019) and recommendation system and advertisement (e.g. Li et al., 2011; Thomas et al., 2017).

Previous success for sequential decision making often requires two key components: (1) a careful design reward function that can provide the supervision signal during learning and (2) an unlimited number of online interactions with the real-world environment (or a carefully designed simulator) to query new unseen region. However, in many scenarios, both components are not allowed. For example, it is hard to define the reward signal in uncountable many extreme situations in autonomous driving; and it is dangerous and risky to directly deploy a learning policy on human to gather information in autonomous medical treatment (Yu et al., 2019). Therefore an *offline* sequential decision making algorithm without reward signal is in demand.

Offline Imitation Learning (IL) offers an elegant way to train intelligent agents for complex task without the knowledge of reward functions or using a simulator. Since the offline imitation learning does not interact with the environment, in order to guide intelligent agents to correct behaviors, it is crucial to have high quality expert demonstrations. The well-known Behavior Cloning (BC) algorithm (Pomerleau, 1988) requires that the demonstrations given for training are all *presumably optimal* and it aims to learn that mapping from state to action from expert demonstration data set.

However in real world scenario, since the demonstration is often collected from human, we cannot guarantee that *all* the demonstrations we collected have high quality. An human expert can make mistakes by accident or due to the hardness of a complicated scenario (e.g., medical diagnosis). Furthermore, even an expert demonstrates a successful behavior, the recorder or the recording system can have a chance to contaminate the data by accident or on purpose (e.g. Eykholt et al., 2018; Neff & Nagy, 2016).

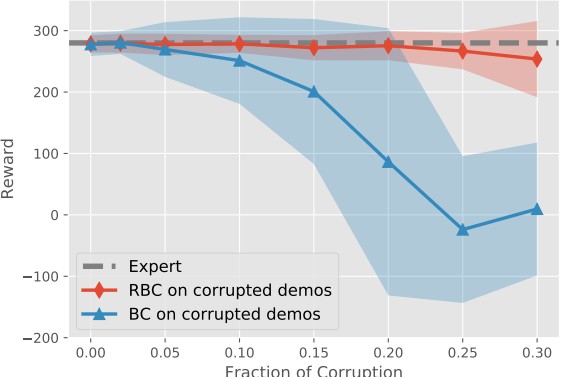

Figure 1: Reward vs. fraction of corruptions in Lunar Lander environment. Shaded region represents one standard deviation for 50 trials. We fix the sample size $N = 4000$ for the demonstration data set, and vary the fraction of corruptions $\epsilon$ up to 30%. Our algorithm Robust Behavior Cloning (RBC) on corrupted demonstrations has nearly the same performance as BC on expert demonstrations (this is the case when $\epsilon = 0$), which achieves expert level. Also, it barely changes when $\epsilon$ grows larger. By contrast, the performance of vanilla BC on corrupted demos fails drastically.

This leads to the central question of the paper:

> Can the optimality assumption on expert demonstrations be weakened or even tolerate arbitrary outliers under offline imitation learning settings?

More concretely, we consider *corrupted demonstrations* setting where the majority of the demonstration data is collected by an expert policy (presumably optimal), and the remaining data can be even arbitrary outliers (the formal definition is presented in Definition 2.1). This has great significance in many applications, such as automated medical diagnosis for healthcare (Yu et al. (2019)) and autonomous driving (Ma et al., 2018), where the historical data (demonstration) is often complicated and noisy which requires robustness consideration.

However, the classical offline imitation learning approaches such as Behavior Cloning (BC) fails drastically under this corrupted demonstration settings. We illustrated this phenomenon in Figure 1. We use BC on a continuous control environment, and the performance of the policy learned by BC drops drastically as the fraction of corruptions increases in the offline demonstration data set. However, our proposed algorithm – Robust Behavior Cloning (Algorithm 1) – is resilient to corruptions in the offline demonstrations. The detailed experimental setup is included in Section 5. We now summarize our contributions as follows.

## 1.1 MAIN CONTRIBUTIONS

- (Algorithm) We consider robustness in offline imitation learning where we have corrupted demonstrations. Our definition for corrupted demonstrations significantly weakens the presumably optimal assumption on demonstration data, and can tolerate a constant $\epsilon$-fraction of state-action pairs to be arbitrarily corrupted. We refer to Definition 2.1 for a more precise statement.

  To deal with this issue, we propose a novel algorithm Robust Behavior Cloning (Algorithm 1) for robust imitation learning. Our algorithm works in the offline setting, without any further interaction with the environment. The core ingredient of our robust algorithm is using a novel median of means objective in policy estimation compared to classical Behavior Cloning. Hence, it's simple to implement, and computationally efficient.

- (Theoretical guarantees) We analyze our Robust Behavior Cloning algorithm when there exists a constant fraction of outliers in the demonstrations under the offline setting. We show that with our RBC achieves nearly the same error scaling and sample complexity compared to vanilla BC with expert

demonstrations. To this end, our algorithm guarantees robustness to corrupted demonstrations at no cost of statistical error. This is the content of Section 4.

- (Empirical support) We validate the predicted robustness and show the effectiveness of our algorithm on different high-dimensional continuous control benchmarks – the vanilla BC is fragile indeed with corrupted demonstrations, and our Robust Behavior Cloning achieves nearly the same performance compared to vanilla BC with expert demonstrations. This is the content of Section 5.

## 2 PROBLEM SETUP

### 2.1 REINFORCEMENT LEARNING AND IMITATION LEARNING

**Markov Decision Process and Reinforcement Learning.** We start the problem setup by introducing the Markov decision process (MDP). An MDP $M = \langle \mathcal{S}, \mathcal{A}, r, \mathsf{P}, \mu_0, \gamma \rangle$ consists of a state space $\mathcal{S}$, an action space $\mathcal{A}$, an unknown reward function $r : \mathcal{S} \times \mathcal{A} \to [0, \mathsf{R}_{\max}]$, an unknown transition kernel $\mathsf{P} : \mathcal{S} \times \mathcal{A} \to \Delta(\mathcal{S})$, an initial state distribution $\mu_0 \in \Delta(\mathcal{S})$, and a discounted factor $\gamma \in (0, 1)$. We use $\Delta$ to denote the probability distributions on the simplex.

An agent acts in a MDP following a policy $\pi(\cdot|\mathbf{s})$, which prescribes a distribution over the action space $\mathcal{A}$ given each state $\mathbf{s} \in \mathcal{S}$. Running the policy starting from the initial distribution $\mathbf{s}_1 \sim \mu_0$ yields a stochastic trajectory $\mathcal{T} := \{\mathbf{s}_t, \mathbf{a}_t, r_t\}_{1 \leq t \leq \infty}$, where $\mathbf{s}_t, \mathbf{a}_t, r_t$ represent the state, action, reward at time $t$ respectively, with $\mathbf{a}_t \sim \pi(\cdot|\mathbf{s}_t)$ and the next state $\mathbf{s}_{t+1}$ follows the unknown transition kernel $\mathbf{s}_{t+1} \sim \mathsf{P}(\cdot|\mathbf{s}_t, \mathbf{a}_t)$. We denote $\rho_{\pi,t} \in \Delta(\mathcal{S} \times \mathcal{A})$ as the marginal joint stationary distribution for state, action at time step $t$, and we define $\rho_\pi = (1 - \gamma) \sum_{i=1}^{\infty} \gamma^t \rho_{\pi,t}$ as visitation distribution for policy $\pi$. For simplicity, we reuse the notation $\rho_\pi(s) = \int_{a \in \mathcal{A}} \rho_\pi(s, a) da$ to denote the marginal distribution over state.

The goal of reinforcement learning is to find the best policy $\pi$ to maximize the expected cumulative return $J_\pi = \mathbb{E}_{\mathcal{T} \sim \pi} \left[ \sum_{i=1}^{\infty} \gamma^t r_t \right]$. Common RL algorithms (e.g., please refer to Szepesvári (2010)) requires online interaction and exploration with the environments. However, this is prohibited in the offline setting.

**Imitation Learning.** Imitation learning (IL) aims to obtain a policy to mimic expert's behavior with demonstration data set $\mathcal{D} = \{(\mathbf{s}_i, \mathbf{a}_i)\}_{i=1}^{N}$ where $N$ is the sample size of $\mathcal{D}$. Note that we do not need any reward signal. Tradition imitation learning assumes perfect (or near-optimal) expert demonstration – for simplification we assume that each state-action pair $(\mathbf{s}_i, \mathbf{a}_i)$ is drawn from the joint stationary distribution of an expert policy $\pi_E$:

$$(\mathbf{s}_i, \mathbf{a}_i) \sim \rho_{\pi_{\mathrm{E}}} \tag{1}$$

Learning from demonstrations with or without online interactions has a long history (e.g., Pomerleau (1988); Ho & Ermon (2016)). The goal of *offline* IL is to learn a policy $\widehat{\pi}^{\mathrm{IL}} = \mathbb{A}(\mathcal{D})$ through an IL algorithm $\mathbb{A}$, given the demonstration data set $\mathcal{D}$, without further interaction with the unknown true transition dynamic $\mathsf{P}$.

**Behavior Cloning.** The Behavior Cloning (BC) is the well known algorithm (Pomerleau, 1988) for IL which only uses offline demonstration data without any interaction with the environment. More specifically, BC solves the following Maximum Likelihood Estimation (MLE) problem, which minimizes the average Negative Log-Likelihood (NLL) for all samples in offline demonstrations $\mathcal{D}$:

$$\widehat{\pi}^{\mathrm{BC}} = \arg\min_{\pi \in \Pi} \frac{1}{N} \sum_{(s,a) \in \mathcal{D}} -\log(\pi(a|s)) \tag{2}$$

Recent works (Agarwal et al., 2019; Rajaraman et al., 2020; Xu et al., 2021) have shown that BC is optimal under the offline setting, and can only be improved with the knowledge of transition dynamic $\mathsf{P}$ *in the worst case*. Also, another line of research considers improving BC with further online interaction of the environment (Brantley et al., 2019) or actively querying an expert (Ross et al., 2011; Ross & Bagnell, 2014).

## 2.2 LEARNING FROM CORRUPTED DEMONSTRATIONS

However, it is sometimes unrealistic to assume that the demonstration data set is collected through a presumably optimal expert policy. In this paper, we propose Definition 2.1 for the corrupted demonstrations, which tolerates gross corruption or model mismatch in offline data set.

**Definition 2.1** (Corrupted Demonstrations). *Let the state-action pair $(\mathbf{s}_i, \mathbf{a}_i)_{i=1}^N$ drawn from the joint stationary distribution of a presumably optimal expert policy $\pi_E$. The corrupted demonstration data $\mathcal{D}$ are generated by the following process: an adversary can choose an arbitrary $\epsilon$-fraction ($\epsilon < 0.5$) of the samples in $[N]$ and modifies them with arbitrary values. We note that $\epsilon$ is a constant independent of the dimensions of the problem. After the corruption, we use $\mathcal{D}$ to denote the corrupted demonstration data set.*

This corruption process can represent gross corruptions or model mismatch in the demonstration data set. To the best of our knowledge, Definition 2.1 is the first definition for corrupted demonstrations in imitation learning which tolerates *arbitrary* corruptions.

In the supervised learning, the well-known Huber's contamination model (Huber (1964)) considers $(\boldsymbol{x}, y) \stackrel{iid}{\sim} (1-\epsilon)P + \epsilon Z$, where $\boldsymbol{x} \in \mathbb{R}^d$ is the explanatory variable (feature) and $y \in \mathbb{R}$ is the response variable. Here, $P$ denotes the *authentic* statistical distribution such as Normal mean estimation or linear regression model, and $Z$ denotes the outliers.

Dealing with corrupted $\boldsymbol{x}$ and $y$ in high dimensions has a long history in the robust statistics community (e.g. Rousseeuw, 1984; Chen et al., 2013; 2017; Yin et al., 2018). However, it's only until recently that robust statistical methods can handle *constant* $\epsilon$-fraction (independent of dimensionality $\mathbb{R}^d$) of outliers in $\boldsymbol{x}$ and $y$ (Klivans et al., 2018; Prasad et al., 2020; Diakonikolas et al., 2019; Liu et al., 2019; 2020; Shen & Sanghavi, 2019; Lugosi & Mendelson, 2019; Lecué & Lerasle, 2020; Jalal et al., 2020). We note that in Imitation Learning, the data collecting process for the demonstrations does not obey i.i.d. assumption in traditional supervised learning due to the temporal dependency.

**Notations.** Throughout this paper, we use $\{c_i\}_{i=1,2,3}$ to denote the universal positive constant. We utilize the big-$O$ notation $f(n) = O(g(n))$ to denote that there exists a positive constant $c_1$ and a natural number $n_0$ such that, for all $n \geq n_0$, we have $f(n) \leq c_1 g(n)$.

## 3 OUR ALGORITHMS

It is well known that the Median-of-Means (MOM) estimator achieves sub-Gaussian concentration bound for one-dimensional mean estimation even though the underlying distribution only has second moment bound (heavy tailed distribution) (interested readers are referred to textbooks such as Nemirovsky & Yudin (1983); Jerrum et al. (1986); Alon et al. (1999)).

The vanilla MOM estimator for one-dimensional mean estimation works like following: (1) randomly partition $N$ samples into $M$ batches; (2) calculates the mean for each batch; (3) outputs the median of these batch mean. Very recently, MOM estimators are used for high dimensional robust regression (Brownlees et al., 2015; Hsu & Sabato, 2016) by applying MOM estimator on the *loss function* of empirical risk minimization process.

### 3.1 ROBUST BEHAVIOR CLONING

Motivated by using MOM estimators on the loss function, we propose Definition 3.1 which uses a MOM objective to handle arbitrary outliers in demonstration data set $(\mathbf{s}, \mathbf{a}) \in \mathcal{D}$.

**Definition 3.1** (Robust Behavior Cloning). *We split the corrupted demonstrations $\mathcal{D}$ into $M$ batches randomly[1]: $\{B_j\}_{j=1}^M$, with the batch size $b \leq \frac{1}{3\epsilon}$. The Robust Behavior Cloning solves the following optimization*

$$\widehat{\pi}^{\text{RBC}} = \arg\min_{\pi \in \Pi} \max_{\pi' \in \Pi} \underset{1 \leq j \leq M}{\text{median}} \left( \ell_j(\pi) - \ell_j(\pi') \right), \tag{3}$$

---

[1]Without loss of generality, we assume that $M$ exactly divides the sample size $N$, and $b = \frac{N}{M}$ is the batch size.

---

**Algorithm 1** Robust Behavior Cloning.

---

1: **Input:** Corrupted demonstrations $\mathcal{D}$
2: **Output:** Robust policy $\widehat{\pi}^{\mathrm{RBC}}$

---

3: Initialize $\pi$ and $\pi'$.
4: **for** $t = 0$ to $T - 1$, **do**
5:  Randomly partition $\mathcal{D}$ to $M$ batches with the batch size $b \leq \frac{1}{3\epsilon}$.
6:  For each batch $j \in [M]$, calculate the loss $\ell_j(\pi) - \ell_j(\pi')$ by eq. (4).
7:  Pick the batch with median loss within $M$ batches

$$\operatorname*{median}_{1 \leq j \leq M} \left( \ell_j(\pi) - \ell_j(\pi') \right),$$

   and evaluate the gradient for $\pi$ and $\pi'$ using back-propagation on that batch
   (i) perform gradient descent on $\pi$.
   (ii) perform gradient ascent on $\pi'$.
8: **end for**
9: **Return:** Robust policy $\widehat{\pi}^{\mathrm{RBC}} = \pi$.

---

*where the loss function $\ell_j(\pi)$ is the average Negative Log-Likelihood in the batch $B_j$:*

$$\ell_j(\pi) = \frac{1}{b} \sum_{(s,a) \in B_j} -\log(\pi(a|s)). \tag{4}$$

The workhorse of Definition 3.1 is eq. (3), which uses a novel variant of Median-of-Means (MOM) tournament procedure (Le Cam (2012); Lugosi & Mendelson (2019); Lecué & Lerasle (2020); Jalal et al. (2020)). In eq. (4), we calculate the average Negative Log-Likelihood (NLL) for a single batch, and $\widehat{\pi}^{\mathrm{RBC}}$ is the solution of a min-max formulation based on the batch loss $\ell_j(\pi)$. Though our algorithm minimizes the robust version of NLL, we do not utilize the traditional iid assumption in the supervised learning.

To gain some intuition of the formulation eq. (3), if we replace the median operator by the mean operator, then RBC is equivalent to BC which just minimizes the empirical average of Negative Log-Likelihood. This is due to the linearity of using the mean operator. However, this is not robust to corrupted demonstration. Hence, we use the median operator on the loss function.

The intuition behind solving this min-max formulation is that the inner variable $\pi'$ needs to get close to $\pi_{\mathrm{E}}$ to maximize the difference of loss function, and the outer variable $\pi$ also need to get close to $\pi_{\mathrm{E}}$. Hence we can guarantee that $\widehat{\pi}^{\mathrm{RBC}}$ will be close to $\pi_{\mathrm{E}}$. In Section 4, we show that under corrupted demonstrations, $\widehat{\pi}^{\mathrm{RBC}}$ in eq. (3) has the same error scaling and sample complexity compared to $\pi_{\mathrm{E}}$.

In Section 4, we provide rigorous statistical guarantees for Definition 3.1. However, the objective function eq. (3) in Definition 3.1 is not convex (in general), hence we use Algorithm 1 as a computational heuristic to solve it.

In each iteration of Algorithm 1, we randomly partition the demonstration data set $\mathcal{D}$ into $M$ batches, and calculate the loss $\ell_j(\pi) - \ell_j(\pi')$ by eq. (4). We then pick the batch $B_{\mathrm{Med}}$ with the median loss, and evaluate the gradient on that batch. We use gradient descent on $\pi$ for the $\arg\min$ part and gradient ascent on $\pi'$ for the $\arg\max$ part. In Section 5, we empirically show that this gradient-based heuristic Algorithm 1 is able to minimize this objective and has good convergence properties. As for the time complexity, when using back-propagation on one batch of samples, our RBC incurs overhead costs compared to vanilla BC, in order to evaluate the loss function for all samples via forward propagation.

# 4 THEORETICAL ANALYSIS

In this section, we provide theoretical guarantees for our RBC algorithm. Since our method (Definition 3.1) directly estimates the conditional probability $\pi(a|s)$ over the offline demonstrations, our theoretical analysis provides guarantees on $\mathbb{E}_{s \sim \rho_{\pi_E}} \left\| \widehat{\pi}^{\text{RBC}}(\cdot|\mathbf{s}) - \pi_E(\cdot|\mathbf{s}) \right\|_{\text{TV}}^2$, which upper bounds the total variation norm compared to $\pi_E$ under the expectation of $s \sim \rho_{\pi_E}$. The ultimate goal of the learned policy is to maximize the expected cumulative return, thus we then provide an upper bound for the sub-optimality $J_{\pi_E} - J_{\widehat{\pi}^{\text{RBC}}}$.

We begin the theoretical analysis by Assumption 4.1, which simplifies our analysis and is common in literature (Agarwal et al., 2019; 2020). By assuming that the policy class $\Pi$ is discrete, our upper bounds depend on the quantity $\log(|\Pi|)/N$, which matches the error rates and sample complexity for using BC with expert demonstrations (Agarwal et al., 2019; 2020).

**Assumption 4.1.** *We assume that the policy class $\Pi$ is discrete, and realizable, i.e., $\pi_E \in \Pi$.*

We first present Theorem 4.1, which shows that minimizing the MOM objective via eq. (3) guarantees the closeness of robust policy to optimal policy in total variation distance.

**Theorem 4.1.** *Suppose we have corrupted demonstration data set $\mathcal{D}$ with sample size $N$ from Definition 2.1, and there exists a constant corruption ratio $\epsilon < 0.5$. Under Assumption 4.1, let $\tau$ to be the output objective value with $\widehat{\pi}^{\text{RBC}}$ in the optimization eq. (3) with the batch size $b \leq \frac{1}{3\epsilon}$, then with probability at least $1 - c_1 \delta$, we have*

$$\mathbb{E}_{s \sim \rho_{\pi_E}} \left\| \widehat{\pi}^{\text{RBC}}(\cdot|\mathbf{s}) - \pi_E(\cdot|\mathbf{s}) \right\|_{\text{TV}}^2 = O \left( \frac{\log(|\Pi|/\delta)}{N} + \tau \right). \tag{5}$$

The proof is collected in Appendix A. We note that the data collection process does not follow the iid assumption, hence we use martingale analysis similar to (Agarwal et al., 2019; 2020). The first part of eq. (5) is the statistical error $\frac{\log(|\Pi|/\delta)}{N}$. The second part is the final objective value in the optimization eq. (3) $\tau$ which includes two parts – the first part scales with $O(\frac{1}{b})$, which is equivalent to the fraction of corruption $O(\epsilon)$. The second part is the sub-optimality gap due to the solving the non-convex optimization.

Our main theorem – Theorem 4.1 – guarantees that a small value of the final objective implies an accurate estimation of policy and hence we can certify estimation quality using the obtained final value of the objective.

Next, we present Theorem 4.2, which guarantees the reward performance of the learned robust policy $\widehat{\pi}^{\text{RBC}}$.

**Theorem 4.2.** *Under the same setting as Theorem 4.1, we have*

$$J_{\pi_E} - J_{\widehat{\pi}^{\text{RBC}}} \leq O \left( \frac{1}{(1-\gamma)^2} \sqrt{\frac{\log(|\Pi|/\delta)}{N} + \tau} \right), \tag{6}$$

*with probability at least $1 - c_1 \delta$.*

The proof is also collected in Appendix A. We note that the error scaling and sample complexity of Theorem 4.1 and Theorem 4.2 matches the vanilla BC with expert demonstrations (Agarwal et al., 2019; 2020).

**Remark 4.1.** *The quadratic dependency on the effective horizon ($\frac{1}{(1-\gamma)^2}$ in the discounted setting or $H^2$ in the episodic setting) is widely known as the compounding error or distribution shift in literature, which is due to the essential limitation of offline imitation learning setting. Recent work (Rajaraman et al., 2020; Xu et al., 2021) shows that this quadratic dependency cannot be improved without any further interaction with the environment or the knowledge of transition dynamic $\mathsf{P}$. Hence BC is actually optimal under no-interaction setting. Also, a line of research considers improving BC by further online interaction with the environment or even active query of the experts (Ross et al., 2011; Brantley et al., 2019; Ross & Bagnell, 2014). Since our work, as a robust counterpart of BC, focuses on the robustness to the corruptions in the offline demonstrations setting, it can be naturally used in online the online setting such as DAGGER (Ross et al., 2011) and Brantley et al. (2019).*

## 5 EXPERIMENTS

In this section, we study the empirical performance of our Robust Behavior Cloning. We evaluate the robustness of Robust Behavior Cloning on several continuous control benchmarks simulated by PyBullet Coumans & Bai (2016) simulator: HopperBulletEnv-v0, Walker2DBulletEnv-v0, HalfCheetahBulletEnv-v0 and AntBulletEnv-v0. Actually, these tasks have true reward function already in the simulator. We will use *only* state observation and action for the imitation algorithm, and we then use the reward to evaluate the obtained policy when running in the simulator.

For each task, we collect the presumably optimal expert trajectories using pre-trained agents from Standard Baselines3[2]. In the experiment, we use Soft Actor-Critic (Haarnoja et al. (2018)) in the Standard Baselines3 pre-trained agents, and we consider it to be an expert.

For the continuous control environments, the action space are bounded. Hence we generate corrupted demonstration data set $\mathcal{D}$ as follows: we first randomly choose $\epsilon$ fraction of samples, and corrupt the action to the boundary (normally $-1$ or $+1$). We note that Definition 2.1 allows for *arbitrary* corruptions, and we choose these outliers' action since it has the maximum effect, and cannot be easily detected.

We compare our RBC algorithm (Algorithm 1) to a number of natural baselines: the first baseline is directly using BC on the corrupted demonstration $\mathcal{D}$ without any robustness consideration. The second one is using BC on the *expert demonstrations* with the same sample size. In different settings, we fix the policy network as 2 hidden layer feed-forward Neural Network of size $\{500, 500\}$ with ReLU activation, which is standard in the baselines.

### 5.1 CONVERGENCE OF OUR ALGORITHM

We illustrate the convergence and the performance of our algorithm to support our theoretical analysis. We track the performance metric of different algorithms vs. epoch number in the whole training process. More specifically, we then evaluate current policy in the simulator for 20 trials, and obtain the mean and standard deviation of cumulative reward for each epoch. This metric corresponds to theoretical bounds in Theorem 4.2.

We focus on four continuous control environments, where the observation space has dimensions around 30, and the action space has boundary $[-1, 1]$. We fix the sample size as 60000, and vary the corruption fraction $\epsilon$ to be $10\%, 20\%$. Figure 2 validates our theory that our Robust Behavior Cloning nearly matches the performance of BC on expert demonstrations for different environments and corruption ratio.

### 5.2 PERFORMANCE UNDER DIFFERENT SETUPS

This is the experiment we have shown in Section 1. In the Lunar Lander control environment, we fix the sample size $N = 4000$, and then vary the fraction of corruptions $\epsilon$. As expected, Figure 1 shows that our RBC is resilient to a constant-fraction of outliers in the demonstrations ranging from 0 to 30%, and it achieves nearly the same performance as the BC on expert demonstrations. In contrast, directly using BC on corrupted demonstrations obtains worse reward performance as the fraction of outliers grows.

## 6 DISCUSSIONS

### 6.1 RELATED WORK

**Imitation Learning.** Behavior Cloning (BC) is the most widely-used imitation learning algorithm (Pomerleau, 1988; Osa et al., 2018) due to its simplicity, effectiveness and scalability, and has been widely used in practice. From a theoretical viewpoint, it has been showed that BC achieves informational optimality in the offline setting (Rajaraman et al., 2020) with *no further online interactions* or the knowledge of the transition dynamic P.

---

[2]The pre-trained agents were cloned from the following repositories: `https://github.com/DLR-RM/stable-baselines3`, `https://github.com/DLR-RM/rl-baselines3-zoo`.

With online interaction, there's a line of research focusing on improving BC in different scenarios – for example, Ross et al. (2011) proposed DAgger (Data Aggregation) by querying the expert policy in the online setting. Brantley et al. (2019) proposed using an ensemble of BC as uncertainty measure and interacts with the environment to improve BC by taking the uncertainty into account, without the need to query the expert. Very recently, (Xu et al., 2021; Rajaraman et al., 2021) leveraged the knowledge of the transition dynamic P to eliminate compounding error/distribution shift issue in BC.

Besides BC, there are other imitation learning algorithms: Ho & Ermon (2016) used generative adversarial networks for distribution matching to learn a reward function; Reddy et al. (2019) provided a reinforcement learning framework to deal with imitation learning by artificially setting the reward; Ghasemipour et al. (2020) unified several existing imitation learning algorithm as minimizing distribution divergence between learned policy and expert demonstration, just to name a few.

**Offline RL.** Reinforcement learning leverages the signal from reward function to train the policy. Different from IL, offline RL often does not require the demonstration to be expert demonstration (e.g. Fujimoto et al., 2019; Fujimoto & Gu, 2021; Kumar et al., 2020) (interested readers are referred to (Levine et al., 2020)), and even expects the offline data with higher coverage for different sub-optimal policies (Buckman et al., 2020; Jin et al., 2021; Rashidinejad et al., 2021). Behavior-agnostic setting (Nachum et al., 2019; Mousavi et al., 2020) even does not require the collected data from a single policy.

The closest relation between offline RL and IL is the learning of stationary visitation distribution, where learning such visitation distribution does not involve with reward signal, similar to IL. A line of recent research especially for off-policy evaluation tries to learn the stationary visitation distribution of a given target policy (e.g. Liu et al., 2018; Nachum et al., 2019; Tang et al., 2020; Mousavi et al., 2020; Dai et al., 2020). Especially Kostrikov et al. (2020) leverages the off-policy evaluation idea to IL area.

**Robustness in IL and RL.** There are several recent papers consider corruption-robust in either RL or IL. In RL, Zhang et al. (2021b) considers that the adversarial corruption may corrupt the whole episode in the online RL setting while a more recent one (Zhang et al., 2021a) considers *offline RL* where $\epsilon$-fraction of the whole data set can be replaced by the outliers. However, the $\epsilon$ dependency scales with the dimension in Zhang et al. (2021a), yet $\epsilon$ can be a constant in this paper for robust offline IL. Many other papers consider perturbations, heavy tails, or corruptions in either reward function (Bubeck et al., 2013) or in transition dynamic (Xu & Mannor, 2012; Tamar et al., 2014; Roy et al., 2017).

The most related papers follow a similar setting of robust IL are (Wu et al., 2019; Tangkaratt et al., 2020; 2021; Brown et al., 2019; Sasaki & Yamashina, 2020), where they consider imperfect or noisy observations in imitation learning. However, their algorithms cannot handle outliers in the demonstrations, and (Wu et al., 2019; Tangkaratt et al., 2020; 2021) require additional *online interactions* with the environment. Our algorithm achieves robustness guarantee from purely *offline* demonstration, without the potentially costly or risky interaction with the real world environment.

## 6.2 SUMMARY AND FUTURE WORKS

In this paper, we considered the corrupted demonstrations issues in imitation learning, and proposed a novel robust algorithm, Robust Behavior Cloning, to deal with the corruptions in offline demonstration data set. The core technique is replacing the vanilla Maximum Likelihood Estimation with a Median-of-Means (MOM) objective which guarantees the policy estimation and reward performance in the presence of constant fraction of outliers. Our algorithm has strong robustness guarantees and works well in practice.

There are several avenues for future work: since our work focuses on the corruption in offline data set, any improvement in *online imitation learning* which utilizes Behavior Cloning would benefit from the corruption-robustness guarantees by our *offline* Robust Behavior Cloning. Also, it would also be of interest to apply our algorithm for real-world environment, such as automated medical diagnosis and autonomous driving.

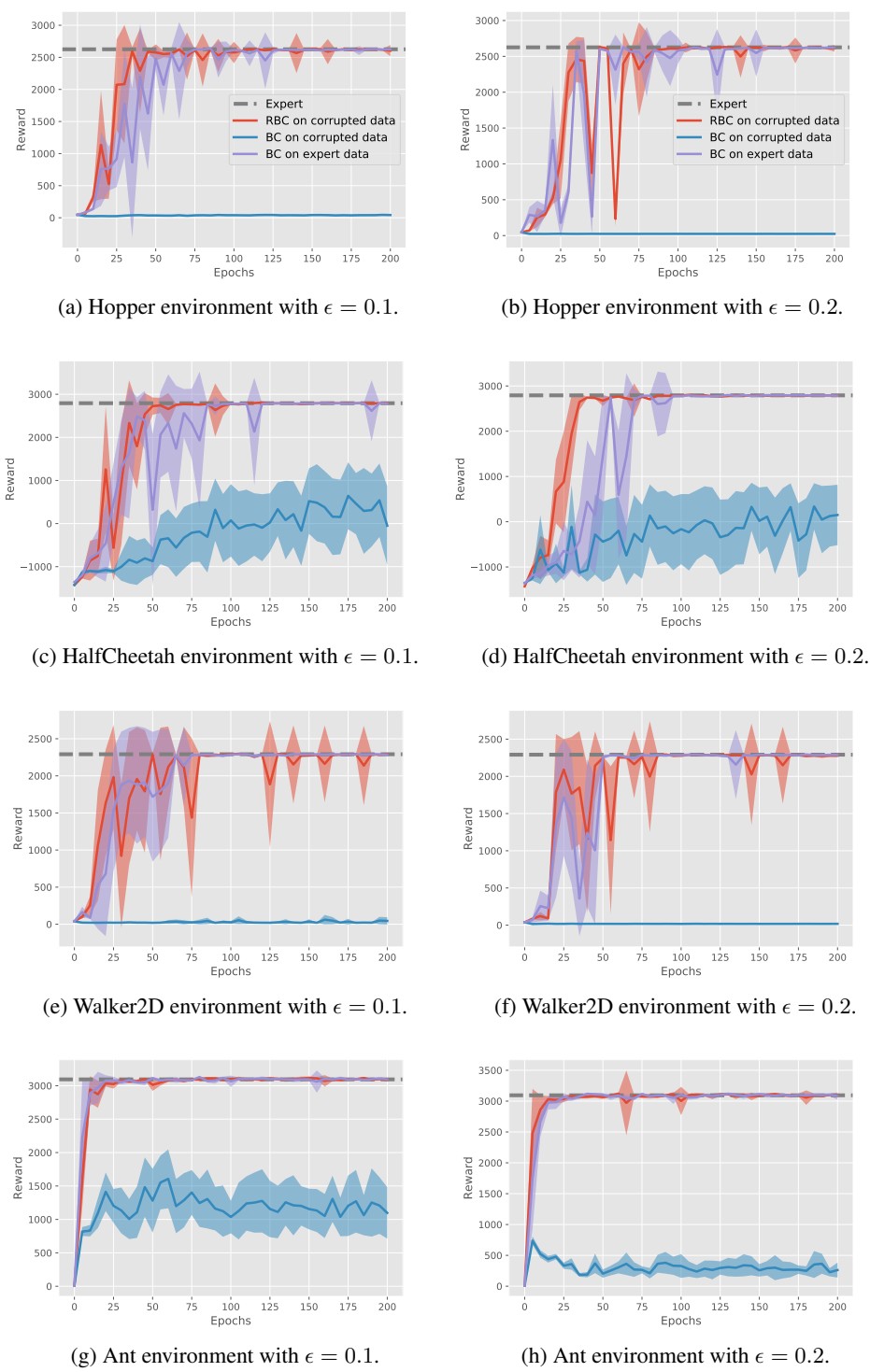

(a) Hopper environment with $\epsilon = 0.1$.

(b) Hopper environment with $\epsilon = 0.2$.

(c) HalfCheetah environment with $\epsilon = 0.1$.

(d) HalfCheetah environment with $\epsilon = 0.2$.

(e) Walker2D environment with $\epsilon = 0.1$.

(f) Walker2D environment with $\epsilon = 0.2$.

(g) Ant environment with $\epsilon = 0.1$.

(h) Ant environment with $\epsilon = 0.2$.

Figure 2: Offline Imitation Learning on four different continuous control tasks with demonstration data of size 60000. We vary the corruption ration $\epsilon = 10\%, 20\%$. For every 5 epochs, we evaluate the current policy in the environment for 20 trials, and the shaded region represents the standard deviation. Vanilla BC on corrupted demonstrations fails to converge to expert policy. Using the robust counterpart Algorithm 1 on corrupted demonstrations has good convergence properties. Surprisingly, our RBC on corrupted demonstrations has nearly the same reward performance of using BC on expert demonstrations.

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

## A    PROOFS

The analysis of maximum likelihood estimation is standard in i.i.d. setting for the supervised learning setting (van de Geer, 2000). In our proofs of the robust offline imitation learning algorithm, the analysis for the sequential decision making leverages the martingale analysis technique from (Zhang, 2006; Agarwal et al., 2020).

Our Robust Behavior Cloning (Definition 3.1) solves the following optimization

$$\widehat{\pi}^{\mathrm{RBC}} = \arg\min_{\pi \in \Pi} \max_{\pi' \in \Pi} \operatorname*{median}_{1 \leq j \leq M} \left( \ell_j(\pi) - \ell_j(\pi') \right), \tag{7}$$

where the loss function $\ell_j(\pi)$ is the average Negative Log-Likelihood in the batch $B_j$:

$$\ell_j(\pi) = \frac{1}{b} \sum_{(\mathbf{s}, \mathbf{a}) \in B_j} - \log(\pi(\mathbf{a}|\mathbf{s})). \tag{8}$$

This can be understood as a robust counterpart for the maximum likelihood estimation in sequential decision process.

With a slight abuse of notation, we use $x_i$ and $y_i$ to denote the observation and action, and the underlying unknown expert distribution is $y_i \sim p(\cdot|x_i)$ and $p(y|x) = f^*(x, y)$. Following Assumption 4.1, we have the realizable $f^* \in \mathcal{F}$, and the discrete function class satisfies $|\mathcal{F}| < \infty$.

Let $\mathcal{D}$ denote the data set and let $\mathcal{D}'$ denote a tangent sequence $\{x_i', y_i'\}_{i=1}^{|\mathcal{D}|}$. The tangent sequence is defined as $x_i' \sim \mathscr{D}_i(x_{1:i-1}, y_{1:i-1})$ and $y_i' \sim p(\cdot|x_i')$. Note here that $x_i'$ follows from the distribution $\mathscr{D}_i$, and depends on the original sequence, hence the tangent sequence is independent conditional on $\mathcal{D}$.

For this martingale process, we first introduce a decoupling Lemma from Agarwal et al. (2020).

**Lemma A.1.** *[Lemma 24 in Agarwal et al. (2020)] Let $\mathcal{D}$ be a dataset, and let $\mathcal{D}'$ be a tangent sequence. Let $\Gamma(f, \mathcal{D}) = \sum_{(x,y) \in \mathcal{D}} \phi(f, (x, y))$ be any function which can be decomposed additively across samples in $\mathcal{D}$. Here, $\phi$ is any function of $f$ and sample $(x, y)$. Let $\widehat{f} = \widehat{f}(\mathcal{D})$ be any estimator taking the dataset $\mathcal{D}$ as input and with range $\mathcal{F}$. Then we have*

$$\mathbb{E}_{\mathcal{D}} \left[ \exp \left( \Gamma(\widehat{f}, \mathcal{D}) - \log \mathbb{E}_{\mathcal{D}'} \exp(\Gamma(\widehat{f}, \mathcal{D}')) - \log |\mathcal{F}| \right) \right] \leq 1.$$

Then we present a Lemma which upper bounds the TV distance via a loss function closely related to KL divergence. Such bounds for probabilistic distributions are discussed extensively in literature such as Tsybakov (2009).

**Lemma A.2.** *[Lemma 25 in Agarwal et al. (2020)] For any two conditional probability densities $f_1, f_2$ and any state distribution $\mathscr{D} \in \Delta(\mathcal{X})$ we have*

$$\mathbb{E}_{x \sim \mathscr{D}} \|f_1(x, \cdot) - f_2(x, \cdot)\|_{\mathrm{TV}}^2 \leq -2 \log \mathbb{E}_{x \sim \mathscr{D}, y \sim f_2(\cdot|x)} \exp \left( -\frac{1}{2} \log \frac{f_2(x, y)}{f_1(x, y)} \right).$$

### A.1    PROOF OF THEOREM 4.1

With these Lemmas in hand, we are now equipped to prove our main theorem (Theorem 4.1), which guarantees the solution $\widehat{\pi}^{\mathrm{RBC}}$ of eq. (3) is close to the optimal policy $\pi_{\mathrm{E}}$ in TV distance.

**Theorem A.1** (Theorem 4.1)**.** *Suppose we have corrupted demonstration data set $\mathcal{D}$ with sample size $N$ from Definition 2.1, and there exists a constant corruption ratio $\epsilon < 0.5$. Under Assump-*

*tion 4.1, let $\tau$ to be the output objective value with $\widehat{\pi}^{\mathrm{RBC}}$ in the optimization eq. (3) with the batch size $b \leq \frac{1}{3\epsilon}$, then with probability at least $1 - c_1\delta$, we have*

$$\mathbb{E}_{s \sim \rho_{\pi_{\mathrm{E}}}} \left\| \widehat{\pi}^{\mathrm{RBC}}(\cdot|\mathbf{s}) - \pi_{\mathrm{E}}(\cdot|\mathbf{s}) \right\|_{\mathrm{TV}}^2 = O\left( \frac{\log(|\Pi|/\delta)}{N} + \tau \right).$$

*Proof of Theorem 4.1.* En route to the proof of Theorem 4.1, we keep using the notations in Lemma A.1 and Lemma A.2, where the state observation is $x$, the action is $y$, and the discrete function class is $\mathcal{F}$.

Similar to Agarwal et al. (2020), we first note that Lemma A.1 can be combined with a simple Chernoff bound to obtain an exponential tail bound. With probability at least $1 - c_1\delta$, we have

$$-\log \mathbb{E}_{\mathcal{D}'} \exp(\Gamma(\widehat{f}, \mathcal{D}')) \leq -\Gamma(\widehat{f}, \mathcal{D}) + \log|\mathcal{F}| + \log(1/\delta). \tag{9}$$

Our proof technique relies on lower bounding the LHS of eq. (9), and upper bounding the RHS eq. (9).

Let the batch size $b \leq \frac{1}{3\epsilon}$, which is a constant in Definition 3.1, then the number of batches $M \geq 3\epsilon N$ such that there exists at least 66% batches without corruptions.

In the definition of RBC (Definition 3.1), we solve

$$\widehat{\pi}^{\mathrm{RBC}} = \arg \min_{\pi \in \Pi} \max_{\pi' \in \Pi} \operatorname{median}_{1 \leq j \leq M} \left( \ell_j(\pi) - \ell_j(\pi') \right). \tag{10}$$

Notice that since $\pi_{\mathrm{E}}$ is one feasible solution of the inner maximization step eq. (10), we can choose $\pi' = \pi_{\mathrm{E}}$. Now we consider the objective function which is the difference of Negative Log-Likelihood between $f$ and $f^*$, i.e., $\ell_j(f) - \ell_j(f^*)$, defined in eq. (4) where

$$\ell_j(\pi) = \frac{1}{b} \sum_{(\mathbf{s},\mathbf{a}) \in B_j} -\log(\pi(\mathbf{a}|\mathbf{s})).$$

Hence, we choose $\Gamma(f, \mathcal{D})$ in Lemma A.1 as

$$\begin{aligned} \Gamma_j(f, \mathcal{D}) &= \frac{N}{b} \sum_{i \in B_j} -\frac{1}{2} \log \frac{f^*(x_i, y_i)}{f(x_i, y_i)} \\ &= \frac{N}{2b} \sum_{i \in B_j} \left( \log f(x_i, y_i) - \log f^*(x_i, y_i) \right), \end{aligned}$$

which is the difference of Negative Log-Likelihood $N(\ell_j(f^*) - \ell_j(f))/2$ evaluated on a single batch $B_j, j \in [M]$. This is actually the objective function on a single batch appeared in eq. (3).

**Lower bound for the LHS of eq. (9).** We apply the concentration bound eq. (9) for such uncorrupted batches, hence the majority of all batches satisfies eq. (9). For those batches, the LHS of

eq. (9) can be lower bounded by the TV distance according to Lemma A.2.

$$
\begin{aligned}
&-\log \mathbb{E}_{\mathcal{D}'}\left[\exp\left(\frac{N}{b}\sum_{i\in B_j}-\frac{1}{2}\log\left(\frac{f^{\star}(x_i',y_i')}{\widehat{f}(x_i',y_i')}\right)\right)\Big|\mathcal{D}\right]\\
&\overset{(i)}{=}-\frac{N}{b}\sum_{i\in B_j}\log\mathbb{E}_{x,y\sim\mathscr{D}_i}\exp\left(-\frac{1}{2}\log\frac{f^{\star}(x,y)}{\widehat{f}(x,y)}\right)\\
&\overset{(ii)}{\geq}\frac{N}{2b}\sum_{i\in B_j}\mathbb{E}_{x\sim\mathscr{D}_i}\left\|\widehat{f}(x,\cdot)-f^{\star}(x,\cdot)\right\|_{\mathrm{TV}}^2,
\end{aligned}
\tag{11}
$$

where (i) follows from the independence between $\widehat{f}$ and $\mathcal{D}'$ due to the decoupling technique, and (ii) follows from Lemma A.2, which is an upper bound of the Total Variation distance.

**Upper bound for the RHS of eq. (9).** Note that the objective is the median of means of each batches and $f^*$ is one feasible solution of the inner maximization step eq. (10). Since $\tau$ is the output objective value with $\widehat{\pi}^{\mathrm{RBC}}$ in the optimization eq. (3), this implies that $\ell_{\mathrm{Med}}(\pi)-\ell_{\mathrm{Med}}(\pi')\leq\tau$ for the median batch $B_{\mathrm{Med}}$, which is equivalent to $-\Gamma_{\mathrm{Med}}(f,\mathcal{D})\leq N\tau/2$.

Hence for the median batch $B_{\mathrm{Med}}$, the RHS of eq. (9) can be upper bounded by

$$
-\Gamma_{\mathrm{Med}}(\widehat{f},\mathcal{D})+\log|\mathcal{F}|+\log(1/\delta)\leq\log|\mathcal{F}|+\log(1/\delta)+N\tau/2.
\tag{12}
$$

Putting together the pieces eq. (11) and eq. (12) for $B_{\mathrm{Med}}$, we have

$$
\mathbb{E}_{s\sim\rho_{\pi_{\mathrm{E}}}}\left\|\widehat{\pi}^{\mathrm{RBC}}(\cdot|s)-\pi_{\mathrm{E}}(\cdot|s)\right\|_{\mathrm{TV}}^2=O\left(\frac{\log(|\mathcal{F}|/\delta)}{N}+\tau\right),
$$

with probability at least $1-c_1\delta$.

$\square$

## A.2 PROOF OF THEOREM 4.2

With the supervised learning guarantees Theorem 4.1 in hand, which provides an upper bound for $\mathbb{E}_{s\sim\rho_{\pi_{\mathrm{E}}}}\left\|\widehat{\pi}^{\mathrm{RBC}}(\cdot|s)-\pi_{\mathrm{E}}(\cdot|s)\right\|_{\mathrm{TV}}^2$, we are now able to present the suboptimality guarantee of the reward for $\widehat{\pi}^{\mathrm{RBC}}$. This bound directly corresponds to the reward performance of a policy.

**Theorem A.2** (Theorem 4.2). *Under the same setting as Theorem 4.1, we have*

$$
J_{\pi_E}-J_{\widehat{\pi}^{\mathrm{RBC}}}\leq O\left(\frac{1}{(1-\gamma)^2}\sqrt{\frac{\log(|\mathcal{F}|/\delta)}{N}+\tau}\right),
$$

*with probability at least $1-c_1\delta$.*

*Proof of Theorem 4.2.* This part is similar to Agarwal et al. (2019), and we have

$$
\begin{aligned}
(1-\gamma)(J_{\pi_E}-J_{\widehat{\pi}^{\mathrm{RBC}}})&=\mathbb{E}_{s\sim\rho_{\pi_{\mathrm{E}}}}\mathbb{E}_{a\sim\pi_{\mathrm{E}}(\cdot|s)}A^{\widehat{\pi}^{\mathrm{RBC}}}(\mathbf{s},\mathbf{a})\\
&\leq\frac{1}{1-\gamma}\sqrt{\mathbb{E}_{s\sim\rho_{\pi_{\mathrm{E}}}}\|\widehat{\pi}^{\mathrm{RBC}}(\cdot|s)-\pi_{\mathrm{E}}(\cdot|s)\|_1^2}\\
&=\frac{2}{1-\gamma}\sqrt{\mathbb{E}_{s\sim\rho_{\pi_{\mathrm{E}}}}\|\widehat{\pi}^{\mathrm{RBC}}(\cdot|s)-\pi_{\mathrm{E}}(\cdot|s)\|_{\mathrm{TV}}^2},
\end{aligned}
$$

where we use the fact that $\sup_{s,a,\pi}|A^\pi(\mathbf{s},\mathbf{a})| \leq \frac{1}{1-\gamma}$ for the advantage function and the reward is always bounded between 0 and 1.

Combining Theorem 4.1, we have

$$J_{\pi_E} - J_{\widehat{\pi}^{\mathrm{RBC}}} \leq O\left(\frac{1}{(1-\gamma)^2}\sqrt{\frac{\log(|\mathcal{F}|/\delta)}{N}} + \tau\right),$$

with probability at least $1 - c_1\delta$. $\qquad\square$

