# OpenReview forum: "Robust Imitation Learning from Corrupted Demonstrations"
_ICLR.cc/2022/Conference — ICLR 2022 Submitted_

### Official Review · Reviewer_9dLk · 2021-10-30

**Correctness:** 2
**Technical Novelty And Significance:** 2
**Empirical Novelty And Significance:** 2
**Recommendation:** 3
**Confidence:** 4

**Main Review:**

### Summary++:

This is quite a difficult paper to judge.  The conception of the scenario and execution within the parameters stated are just about fine.  A lot of imitation learning papers do assume that you can query the environment or an expert policy, which is not always feasible.  This paper uses strictly just demonstrations, which is interesting, and I don’t think gets enough attention.  The core of the learning method is essentially gradient descent on a dynamically selected subset of each minibatch, and so while there isn’t much wrong there, the innovation is limited.

What I believe to be the “secret sauce” is basically uncommended on (see A), and the absolute contribution is low as it basically combines existing components in a somewhat contrived scenario that essentially fits that combination of components (see B).  I also have problems with the formal statements and math (see C, D, E).  I believe there is also a tension between the motivation for the method and the method itself (see F).  Finally, the empirical validation is chronically lacking for such a simple and broadly applicable method (see G).

### Major Comments:

A - The method itself is incredibly simple, which in a way is to be commended, I suppose.  However, the crux of the method, in my eyes, is that backpropogating through only the median loss term just discards the outliers (as intended), such that those samples are never used in a gradient computation.  This is quite a succinct method for “automated outlier detection,” akin to something like RANSAC for localization with outliers.  This is a nice intuition and application of this idea, but it is dressed up to be way more complex than this in the paper.  When the method is stripped back to this, it is a notably less significant advance.  I also think this perspective engenders analysis and discussion that is currently lacking from the paper.

B - The corrupted samples are selected at random from the demonstrations, and, the ratio of corrupted to non-corrupted is quite low (<10%).  Therefore, it is perfectly reasonable that the MoM operation will discard these samples.  It is unclear to me what the effect would be with higher percentages of corrupted samples.  I expect that the gradient estimators would become *very* high-variance as the probability of each small minibatch having at least a single outlier tends to one.  (This relatively simple but hyper-instructive mathematical analysis is also absent from the paper).  The corrupted samples are drawn with uniform probability, and so there is correct supervision present in the corrupted dataset, it is just a case of filtering out the bad samples. In this sense, this whole paper is just defining a type of filter, which is less significant as a methodological innovation again.  I think something like a semi-supervised method learning attaching a Bernoulli distributed random mask on which samples are outliers, or any unsupervised outlier detection/clustering method, would be much more scalable, performant, and appropriate for higher corruption ratios.

B.1 - The corruptions are defined by driving the action to the extrema of the allowable range.  This is the worst-case scenario for the B.C. baseline, and the best-case scenario for the proposed method.  This is also quite a rare case, as the corruption would likely be biased, but not _extreme_.  I therefore believe that the comparison is unfair, to the point of being invalid.

C - I seriously question the structure of the objective in Eq. (3).  I cannot see how the introduction of a second policy $\pi’$ affects this optimization.  As both $\pi$ and $\pi’$ are defined on the same space, I _think_ this expression reduces to simply:

\begin{equation}
= argmin_{\pi \in \Pi} 2 * l_j (\pi) = argmin_{\pi \in \Pi} l_j (\pi)
\end{equation}

(you can essentially change the max to a min and change the minus to a plus, or, (i think) the loss function for the difference is symmetric (up to a minus) in $\pi$ and $\pi’$ and so the solution must be at $\pi = \pi’$. )

I also think there is an expectation over the minibatch partitions missing from the objective.

D - The “theory” sections add nothing to this paper, as far as i’m concerned.  I believe they serve no purpose other than making the paper seem more significant/complex than it is.  _Especially_ given that the only bit of theoretical analysis I see as required -- the probability that any given minibatch is uncorrupted, and hence provides unbiased gradient signal -- is not included.

E - The definition of a corrupted dataset is also not entirely consistent with the definition used later on.  You discuss an “adversary” getting to change the observed action for a chosen \epsilon-subset of the data, but, later on, you then randomly sample the points to corrupt and set them to an extreme+random value.  This is quite far apart from the idea of a hypothetical adversary intentionally corrupting the dataset.  As I said in B, the correct supervision is in the data in your setup, whereas in the real world, samples may be consistently biased in regions of subspace.  This means that the expert policy could not be recovered, and hence I am deeply skeptical of your “guarantees” that the expert policy will be recovered.

F - In the introduction, the authors (in-part) motivate the problem of learning an agent as being hindered by the inability to conduct experiments in hazardous or dangerous regions of state space.

F.1 - Behavioural cloning requires samples in all parts of visited state space to ensure performance, and therefore having missing regions of state space or biased demonstrations in these regions of state space can lead to arbitrarily sub-optimal results.  Simulation-based methods are the _only_ way to ensure consistency.

F.2 - The authors also comment on the difficulty of specifying a reward function.  However, IL still requires you to specify a loss function, which may or may not produce results that are optimal in terms of reward, and hence IL/BC does not alleviate this either. Minimizing a loss term does not imply optimality in reward space (i.e. [Warrington et al., 2020]).

F.3 - Limiting yourself to not use environment interactions is interesting, but I feel that this is a critical restriction that ultimately limits the utility of this method.  I struggle to see a scenario where demonstrations are too few and too corrupted for B.C. to be viable even with manual data cleaning, but where application of the proposed method would enable practitioners to be like “sure, let’s use these highly corrupted demonstrations anyway”...  That may be a personal thing, but I feel like this method may struggle to find impact anywhere, and certainly with the broad and safety-critical purview motivated in the introduction.  To be convinced of this, I would need to see this method crack a previously intractable and real-world task.

G - The empirical validation is far too thin in my opinion.  Testing on two (simple) gym environments is not especially challenging, and, as alluded to above, the rate of corruption is pretty low and corrupted samples are easily detectable.  The baselines compared to are pretty much the most simple thing one could have done, and while I am not overly familiar with B.C. methods, I struggle to believe that nothing more complex than L-2 regression has been done prior to this.  Again, as alluded to above, I think semi-supervised methods (or even historic methods such as Jackknife sampling) would easily outperform this method in more complex settings.  I understand that the highly restricted scenario limits the diversity of methods that can be use, but I do not believe that this method has been “battle tested” when it is compared against baselines that are either (a) infeasible or (b) known/designed to fail.

H - I do take issue with the use of “imitation learning” throughout -- you are explicitly doing the more restricted “behavioural cloning”, as opposed to the slightly more general family of imitation learning methods.


### Minor Comments:
1 - Only proper nouns should be capitalized: i.e. “Robust Behavior Cloning” -> “robust behavior cloning”;  “Negative Log-Likelihood” -> “negative log-likelihood”.

2 - Avoid colloquial language and contractions: i.e. “demos”.

3 - The figures are massive for how much information is conveyed in them.

4 - The math is pretty impenetrable.


Warrington, et al. "Robust asymmetric learning in pomdps." International Conference on Machine Learning. PMLR, 2021.


**Summary Of The Paper:**

This paper tackles behavioural cloning from expert demonstrations, where a fixed fraction of the demonstrations are corrupted.  This is a somewhat (see comments below) practical scenario where trajectories gathered are limited or biased in response to constraints on collection.  A crucial (again, see comments below) point is that _no_ further environment interactions are allowed -- only the possibly corrupted demonstrations can be used.  The paper provides a definition of a corrupted dataset, an objective for training using a corrupted dataset, and provides some convergence and error bounds.  The method is tested on two standard Gym environments, and some ablation studies are provided.

**Summary Of The Review:**

The paper itself is not especially well written, having numerous typos and grammatical mistakes, and is evaluated on the barest minimum of tasks (two relatively simple gym environments), with no convincing baselines or alternative methods compared to.  I am also currently unconvinced by the core objective (eq. (3)), and the relevance of the math presented, although I invite the authors to justify these both to me.  Finally, I think the setup itself is both too restrictive to have significant practical impact, and is actually in conflict with the original motivation for the work.

Overall, I do not think this paper is of the requisite quality for acceptance at this point in time.  I encourage the authors to explore removing some of the inherent restrictions on the method, validating the method on a wider range of environments, seek out alternative methods to compare to, and to submit this work to a workshop to further develop the work and obtain feedback.

---

> ### Author Response · Authors · 2021-11-23
> **Thank you for your feedback**
>
> We thank the reviewer for the feedback. Regarding your concerns, we want to make several clarifications:
>
> A -- Using backpropogation through the median loss term is the computational heuristic for solving the Robust Behavior Cloning method. It is highly nontrivial that we show our RBC is robust up to constant fraction of outliers in the high dimensional observations space, theoretically and experimentally. Previous methods such as RANSAC DO NOT have such robustness guarantees.
> We do provide intuitions and rigorous analysis for proposing the RBC objective, and leverage a gradient-based method to solve it.
>
> B -- In the revised submission, we extend our experiments to four much-harder high dimensional continous control task. Both theory and experiments validate that our Robust Behavior Cloning can handle constant fraction of outliers, as long as the batch size $b\leq \frac{1}{3\epsilon}$.
>
> C -- We can only use the "reduction" to one parameter \pi when we minimize the mean of Negative Log-Likelihood. This is due to the linearity of using the mean operator. However, the reduction statemetn is not true in the median case.
>
> D -- Our theoretical results are highly non-trivial since we show our RBC is robust up to constant (independent of dimension) fraction of outliers in the high dimensional observations space. Just showing unbiased gradient signal cannot lead to the correct scaling as in our theorems, and our proof technique requires concentration bounds for a majority of batches.
>
> E -- We propose the corrupted demonstration definition, where a constant fraction of samples are arbitrarily corrupted. Certainly, this cannot deal ALL the case in the real world. It is worthwhile to consider another robust model for the case where samples may be consistently biased in regions of subspace.
>
> F -- We proposed a method for Offline Imitation Learning setting, where we cannot interact with the environment and we have no reward signal. Behavior Cloning has proved to be optimal in this setting when we have perfect expert demonstrations. Dealing with online setting or equipped with a reward signal is certainly interesting, but it's beyond the scope of this paper.
>
> G -- In the revised submission, we extend our experiments to four much-harder high dimensional continous control task. Both theory and experiments validate that our Robust Behavior Cloning can handle constant fraction of outliers, as long as the batch size $b\leq \frac{1}{3\epsilon}$. Also, there's a line of research dealing with noisy observation in the offline imitation learning setting, but our corrutped setting is much harder than the noisy setting. We will add comparisions in the future revisions.
>
> H -- We thank the constructive feedback, and we have updated them in the revised submission.

---

> > ### Comment · Reviewer_9dLk · 2021-11-30
> > **Response**
> >
> > Dear Authors,
> >
> > Thank you for your response.  Thank you for adding additional experiments and clarifications to the paper, these have indeed strengthened the paper.
> >
> > You have however also updated some of the prose which I fundamentally do not agree with. For instance, you have added: "We note that Definition 2.1 allows for arbitrary corruptions, and we choose these outliers’ action since it has the maximum effect, and cannot be easily detected." which directly contravenes one of my comments and another reviewers comment, and is incorrect, and, I believe, actively misleading to the reader.
> >
> > While my stance on this paper has softened slightly since I first read it, I still stand by the majority of my initial comments, and believe that it is not of publication quality yet.  I will therefore not be altering my review scores.  Good luck going forwards.  9dLk.

---

### Official Review · Reviewer_Sr3Z · 2021-11-01

**Correctness:** 3
**Technical Novelty And Significance:** 2
**Empirical Novelty And Significance:** 3
**Recommendation:** 5
**Confidence:** 3

**Main Review:**

Strengths:

+ The paper considers a problem of relevance to the ICLR community, and proposes what appears to be a novel algorithm with both empirical and theoretical support.

+ I found the paper very clear and easy to read

+ The theoretical results are tight, in that they match the optimal statistical rates achievable in offline IL.

Weaknesses:

- Theorems 4.1 and 4.2 have no explicit dependence on the fraction of corrupted data points $\varepsilon$ and/or the number of batches $M$ needed to ensure that at least 50% of the batches are not corrupted (which is assumed in the proof).  Presumably there is a tradeoff/dependence here.  In the extreme case where $\varepsilon\approx .5$, it would seem that each "batch" would be of size 1, at which point it isn't clear if/how the algorithm would work.  Some more explicit delineations of these constants would be very useful.

- The empirical evaluation was somewhat underwhelming: I would have expected to see several more environments tested.

- No comparison to other robust imitation learning algorithms were provided, and so it is unclear if the algorithm is doing well, or if the tasks are easy enough that any sensible approach to limiting the effects of outliers would do well.  If the reason is that no other offline robust IL algorithms exist, then why not compare DAGGER w/ a MOM cost (which the authors repeatedly state is easy to implement) to some of the interactive robust IL methods outlined in the related work section.  At the very least, comparisons to more sophisticated offline IL methods should be done, e.g., the method proposed in Ho and Ermon 2016.  Statistical optimality and practical performance are not always the same, and so only comparing to BC is not as strong as suggested by the results as Agarwal et al.



**Summary Of The Paper:**

This paper considers an offline imitation learning task wherein a constant $\varepsilon$-fraction of demonstrations have been potentially arbitrarily corrupted.  This data corruption is motivated by sub-optimal experts and/or erroneous/adversarial sensors.  In order to address this, the paper proposes minimizing a Median-of-Means (MOM) objective.  Theoretically, it is shown that up to constant factors, the resulting learned policy achieves the same statistical rates as behavior cloning, which was shown to be optimal for the offline imitation learning setting in recent work by Agarwal et al.  This analysis is done under the assumption of a discrete policy class containing the expert policy.  Algorithmically, an approach based on alternating gradient descent/ascent is proposed, and it is shown on two continuous control benchmarks in PyBullet (LunarLanderContinuous-v2 and HalfCheetahBulletEnv0) that the proposed approach compares favorably to a behavior cloning as applied to uncorrupted data, whereas vanilla behavior cloning performs very poorly.

**Summary Of The Review:**

This paper presents a novel and in my perspective interesting approach to tackling IL in the corrupted data setting.  However, I believe it would benefit from a refinement of the theoretical results to highlight the role that the fraction $\varepsilon$ of corrputed data plays, as well as from more extensive empirical evaluation, as described above.

---

> ### Author Response · Authors · 2021-11-23
> **Thank you for your feedback**
>
> We thank the reviewer for the feedback. Regarding your concerns, we want to make several clarifications:
>
> In the revised submission, we extend our experiments to four much-harder continous control task. Both theory and experiments validate that our Robust Behavior Cloning can handle constant fraction of outliers ($\epsilon < 0.5$), as long as the batch size  $b \leq \frac{1}{3\epsilon}$. The upper bound in Theorem 4.1 and 4.2 explicitely dependes on $\tau$, which is the final objective value in the optimization of our Robust Behavior Cloning. This final objective value consists of $O(\epsilon)$ and the sub-optimality gap due to the non-convex nature of the problem.
>
> We also thank the reviewer for pointing out the references. To the best of our knowledge, there is no clear definition for the corruptions in the online interaction with the MDP environment, hence we do not evaluate our algorithm in the online setting. We will add comparisons to existing offline imitation learning methods such as [1] in further revisions.
>
> [1] Sasaki, Fumihiro, and Ryota Yamashina. "Behavioral Cloning from Noisy Demonstrations." ICLR. 2021.

---

### Official Review · Reviewer_rs1D · 2021-11-01

**Correctness:** 2
**Technical Novelty And Significance:** 3
**Empirical Novelty And Significance:** 2
**Recommendation:** 3
**Confidence:** 4

**Main Review:**

Strength:

1.	The proposed MOM objective for robust imitation learning is novel.
2.	Theoretical justifications are available showing the validity of the proposed RBC.
3.	The paper is well-organized and clearly written.

Weakness:

1.	In the experiments, data corruption changes a sampled action to a boundary value which does not match exactly with the demonstration corruption definition in Def. 2.1. It would be better for the authors to corrupt the actions with an arbitrary value within the action value range.

2.	The experiments focus on tasks with low-dimensional state spaces. Considering that BC also achieves good performance in tasks like Ant, it would be good for the authors to consider tasks with higher-dimensional state spaces. Moreover, there are only two tasks in the experiment section, which does not show a strong validation of the effectiveness of the proposed solution.

3.	The performance of RBC is not convincing in the HalfCheetah task as only 1% of corruption is added. It would be better for the authors to show RBC’s performance on HalfCheetah when the corruption rate increases to 20%, to better show the advantage of the proposed RBC.

4.	Only behavior cloning is compared in the experiment section. Baselines that assume expert demonstrations such as EDM[1] and DFSN[2] should also be compared to show the robustness to data corruption and better show the necessity of the proposed solution.

[1] Jarrett, D., Bica, I., & van der Schaar, M. (2020). Strictly batch imitation learning by energy-based distribution matching. arXiv preprint arXiv:2006.14154.
[2] DonghunLee,SrivatsanSrinivasan,andFinaleDoshi-Velez.Trulybatchapprenticeshiplearning with deep successor features. International Joint Conference on Artificial Intelligence (IJCAI), 2019.


**Summary Of The Paper:**

In this paper, the authors aim to find an offline solution to the imitation learning problem with corrupted demonstration data. They propose a simple robust behavior cloning (RBC) approach based on the Median-of-Means (MOM) objective. The contributions of this work are as follows:

1.	A novel RBC algorithm is proposed with a MOM objective in policy estimation based on behavior cloning.
2.	The authors provided theoretical justifications on the error scaling and sample complexity of the proposed RBC to show that RBC guarantees robustness to corrupted demonstrations at no cost of statistical error.
3.	Experiments on some low-dimensional tasks show the effectiveness of the proposed RBC.

**Summary Of The Review:**

In general, the authors propose a novel and interesting MOM objective for robust imitation learning from corrupted data. However, there are some weaknesses (listed above), including no sufficient experiment results supporting the effectiveness of the proposed solution, preventing me from giving an acceptance.

---

> ### Author Response · Authors · 2021-11-23
> **Thank you for your feedback**
>
> We thank the reviewer for the feedback. Regarding your concerns, we want to make several clarifications:
>
> In our experiments, we adversarially choose the outliers at the boudary of the action space. This will have the biggest impact to our robust algorithm. We will add different set up to further revisions.
>
> We thank the reviewer for the constructive advice. In the revised submission, we extend our experiments to four much-harder continous control task. Both theory and experiments validate that our Robust Behavior Cloning can handle constant $\epsilon$-fraction of outliers ($\epsilon < 0.5$) in the high dimensions, as long as the batch size $b \leq \frac{1}{3 \epsilon}$.
>
> We also thank the reviewer for pointing out thesereferences.We note that the corrupted demonstration setting is much harder, and we will add the comparisons to further revisions.

---

### Official Review · Reviewer_tcU1 · 2021-11-03

**Correctness:** 4
**Technical Novelty And Significance:** 3
**Empirical Novelty And Significance:** 2
**Recommendation:** 5
**Confidence:** 3

**Main Review:**

Strength:

The authors provide theoretical and empirical analysis that the proposed algorithm is simple yet effective.

Weakness:

In the theorems, the authors assume that a small enough $\epsilon$ exists. However, with very small $\epsilon$, BC also performs well. It will be a stronger statement when the $\epsilon$ is included in the bounds, not assumption.

I think the previous work [1] is very similar to this work. This paper also proposes a robust (offline) IL algorithm from the noisy expert demonstration. Because this work is applicable in the offline, some discussions or empirical comparisons should be provided.

[1] Sasaki, Fumihiro, and Ryota Yamashina. "Behavioral Cloning from Noisy Demonstrations." ICLR. 2021.


**Summary Of The Paper:**

This paper proposes the definition of corrupted demonstrations and a new robust algorithm for offline imitation learning from corrupted demonstrations. The main idea to guarantee the robustness is median-of-means (MOM), which is used for high-dimensional robust regression.
The authors propose an optimization formula based on MOM. The proposed algorithm, named robust behavior cloning (RBC), solves the min-max-median optimization to train two policies $\pi$ and $\pi’$.

Under the assumption that the policy class is discrete, two meaningful theorems are provided. The first theorem shows that the policy obtained from RBC is close to the expert policy with high probability. The second theorem shows the degree of suboptimality measured by the difference between the expected return of the policy obtained from RBC and the expert policy.

Finally, the authors provide empirical results on two continuous control benchmarks. In both domains, the proposed algorithm achieves competitive performance with BC on expert demos and outperforms BC on corrupted demonstrations by a large margin.


**Summary Of The Review:**

At least one highly related work is missed and the statements of the theorem require small $\epsilon$.

However, I think the proposed algorithm based on MOM and theoretical analysis are interesting.

---

> ### Author Response · Authors · 2021-11-23
> **Thank you for your feedback**
>
> We thank the reviewer for the feedback. Regarding your concerns, we want to make several clarifications:
>
> In the revised submission, we extend our experiments to four much-harder continous control task. Both theory and experiments validate that our Robust Behavior Cloning can handle constant $\epsilon$-fraction of outliers ($\epsilon < 0.5$), as long as the batch size $b \leq \frac{1}{3 \epsilon}$. Also, our experiments show that BC fails drastically even when $\epsilon$ is small, yet our RBC has predicted results.
>
> We also thank the reviewer for pointing out the references. Though it’s worthwhile to compare with the noisy BC paper, the noisy demonstration setting is much weaker than the corrupted setting, and we will add the comparisons to further revisions.

---

### Decision · Program_Chairs · 2022-01-20

**Decision:**

Reject

**Comment:**

This submission addressed offline imitation learning problem with non-optimal demonstrations. The AC went through the draft, reviews, and replies. The AC agrees with all reviewers that the mathematical analysis, empirical evaluation, and general quality of writing haven't reached the bar of ICLR papers.